# Relationship Among Body Mass Index, Physical Activity, Sedentary Behavior, and Blood Pressure in Portuguese Children and Adolescents: A Cross-Sectional Study

**DOI:** 10.3390/ijerph22010020

**Published:** 2024-12-28

**Authors:** Edmar Mendes, Paulo Farinatti, Alynne Andaki, André Pereira dos Santos, Jéssica Cordeiro, Susana Vale, Andreia Pizarro, Maria Paula Santos, Jorge Mota

**Affiliations:** 1Department of Sports Sciences, Federal University of Triângulo Mineiro, Uberaba 38025-350, MG, Brazil; alynne.andaki@uftm.edu.br; 2Research Centre of Physical Activity, Health, and Leisure, Laboratory for Integrative and Translational Research in Population Health (ITR), Faculty of Sports, University of Porto, 4200-450 Porto, Portugal; paulo.farinatti@uerj.br (P.F.); andrepereira.educa@gmail.com (A.P.d.S.); jessica.cordeiro@alumni.usp.br (J.C.); susanavale@ese.ipp.pt (S.V.); anpizarro@fade.up.pt (A.P.); msantos@fade.up.pt (M.P.S.); jmota@fade.up.pt (J.M.); 3Laboratory of Physical Activity and Health Promotion, Institute of Physical Education and Sports, State University of Rio de Janeiro, Rio de Janeiro 20550-013, RJ, Brazil; 4College of Nursing of Ribeirão Preto, University of São Paulo, Ribeirão Preto 14040-902, SP, Brazil; 5School of Physical Education and Sport of Ribeirão Preto, University of São Paulo, Ribeirão Preto 14040-900, SP, Brazil; 6Study and Research Group in Anthropometry, Training and Sport, (GEPEATE), University of São Paulo, Ribeirão Preto 14040-900, SP, Brazil; 7Department of Child, Family and Population Health Nursing, University of Washington, Seattle, WA 351202, USA

**Keywords:** pediatric hypertension, childhood obesity, waist-to-height ratio, cardiovascular risk, health

## Abstract

This study examined the associations between age, adiposity, physical activity, sedentary behavior, and elevated blood pressure (BP) in 2901 Portuguese children and adolescents aged 2–18. BP, body mass index (BMI), waist-to-height ratio (WHtR), physical activity, and sedentary behavior were measured. Elevated BP was defined as a BP above the 90th percentile for age, sex, and height. Multivariable analyses assessed the prevalence ratios (PR) of elevated BP across age groups, BMI, WHtR, physical activity, and sedentary behavior. Results showed that older age, especially among middle schoolers, was associated with a 1.8 times higher risk of elevated BP than preschoolers. Participants with a higher BMI (overweight/obese) and a WHtR ≥ 0.50 had a 1.49- and 1.4-times higher risk of elevated BP, respectively. Those who did not meet the recommended 60 min of moderate-to-vigorous physical activity (MVPA) per day showed a 1.63 times higher risk of elevated BP, whereas the association between sedentary behavior and BP was not significant after adjustment. These findings highlight age, higher BMI, central obesity, and insufficient physical activity as key factors associated with elevated BP, underscoring the need for early monitoring and intervention to prevent hypertension in this population.

## 1. Introduction

Hypertension, traditionally viewed as a chronic condition mainly affecting adults, has seen a worrying increase among adolescents and even children. This trend can be attributed in part to significant lifestyle changes in recent years. Increased consumption of processed, high-calorie foods, more sedentary behaviors due to screen-based entertainment, such as video games, and a growing preference for indoor activities over outdoor physical play have all contributed to the rising prevalence of lifestyle-related diseases in this age group, including hypertension [1].

Although extensive research has examined the prevalence of hypertension in children and adolescents [2,3], comprehensive global estimates remain limited. A systematic review and meta-analysis involving 47 studies aimed at assessing the global prevalence of pediatric hypertension found that an increase in childhood hypertension between 1994 and 2018 was associated with a higher body mass index (BMI). The combined estimate indicated a 4.0% prevalence among individuals aged ≤ 19 years. In 2015, childhood hypertension prevalence ranged from 4.3% in 6-year-olds to 3.3% in 19-year-olds, with the highest rate of 7.9% observed among 14-year-olds [4].

Nutritional status and adiposity play a significant role in pediatric hypertension. Children with an increased BMI are more than three times more likely to have elevated blood pressure (BP) than those with normal weight [5]. A 1 kg/m^2^ increase in BMI is linked to a 1.4 mmHg rise in systolic BP among prepubertal children [6]. Even within the normal range, higher BMI percentiles correlate with a greater risk of elevated BP in children and adolescents [7]. Furthermore, studies indicate that children with a higher BMI or BP tend to show greater arterial stiffness than those with values in the normal range [8].

Physical inactivity is a major factor in increased cardiovascular risk during childhood [9]. Approximately 80% of children do not meet the recommended 60 min of moderate-to-vigorous physical activity (MVPA) per day [10]. Prolonged sedentary time, largely due to screen use, is also on the rise, with children aged 10–12 spending around 60% of their waking hours in sedentary activities [11,12]. Physical activity is essential for child development and for reducing cardiovascular risk [8]. Evidence suggests that lifestyle changes, particularly increased physical activity, are the most effective non-pharmacological methods for lowering BP [13].

Recent studies emphasize the importance of physical activity in reducing cardiovascular risk and hypertension in children and adolescents. A systematic review and network meta-analysis [14] found that physical activity interventions significantly lower both systolic and diastolic BP, with greater effects when combined with nutrition and behavior modification. Similarly, a dose–response meta-analysis [15] showed a positive association between screen time and hypertension, with hypertensive children having 0.79 h more screen time than normotensive children. These findings highlight the need for further research on modifiable lifestyle factors affecting BP in youth.

Identifying the factors associated with elevated BP in children and adolescents is crucial, as hypertension is a primary cardiovascular risk factor linked to morbidity and mortality. Adiposity, nutritional status, physical activity, and sedentary behavior are key components of cardiovascular health during childhood. In this context, the present cross-sectional study aimed to explore the relationships between adiposity, physical activity, sedentary behavior, and age-related BP in a large cohort of Portuguese children and adolescents. Based on existing evidence, we hypothesized that higher levels of adiposity and sedentary behavior would be positively associated with elevated BP, while greater physical activity would show an inverse association with BP levels.

## 2. Materials and Methods

This cross-sectional study complied with the STROBE (Strengthening the Reporting of Observational Studies in Epidemiology) guidelines [16]. We selected a non-randomized sample of 2901 children and adolescents (1469 boys) from three databases: (a) Preschool Physical Activity, Body Composition, and Lifestyle Study (PRESTYLE), (b) International Physical Activity Environment Network (IPEN Adolescent), and (c) Environmental Support for Leisure and Active Transport (SALTA Project). Participants were recruited from local kindergartens, primary schools, and secondary schools located in the metropolitan area of Porto, Portugal. Inclusion criteria involved children attending kindergarten (for the PRESTYLE study), primary school (for the SALTA Project), and secondary school (for the IPEN Adolescent study). Exclusion criteria included significant health conditions that could impair participation in physical activity or body composition assessments. Baseline data were collected from the PRESTYLE study between 2013–2015, the IPEN Adolescent study between 2010–2011, and the SALTA Project between 2013–2014.

The participants were categorized into four age groups: preschoolers (2–5 years), primary schoolers (6–10 years), middle schoolers (11–14 years), and high schoolers (15–18 years). These groups were defined based on studies indicating behavioral transitions among children and adolescents of different age ranges [17,18,19]. Recruitment took place in educational institutions, ranging from kindergartens to high schools in the metropolitan area of Porto, Portugal. The Ethics Committee of the Faculty of Sport at the University of Porto approved the studies, and their implementation complied with the recommendations in the Declaration of Helsinki. Written informed consent was obtained from all parents.

Body mass was assessed using a digital electronic scale (TanitaTM Inner Scan BC 532, Tanita Corporation, Tokyo, Japan). Height was measured using a portable Holtain stadiometer (HoltainTM Limited, Crymych, UK). BMI was calculated (Kg/m^2^) and categorized according to the BMI cut-offs of the World Health Organization [20]. For children aged 2–5 years, normal weight was defined as BMI ≥ −2 standard deviations (SD) to <+2 SD, overweight as ≥+2 SD to <+3 SD, and obesity as ≥+3 SD. For those aged 5–18 years, normal weight was ≥−2 SD to <+1 SD, overweight ≥ +1 SD to <+2 SD, and obesity ≥ +2 SD. WC was measured at the midpoint between the last costal arch and iliac crest, and the hip circumference at the largest pelvic area. Waist-to-height ratio (WHtR) was calculated as the quotient between WC (cm) and height (cm).

After 5 min of seated rest, BP was measured on the right arm using a Colin BP 8800 device (CritikronTM, Inc., Tampa, FL, USA). An average of two measurements for systolic blood pressure (SBP) and diastolic blood pressure (DBP) were recorded. A third measurement was taken if the difference between the two measurements was greater than 2 mmHg. All BP measurements were conducted between 8:00 and 11:00 by the same investigator using the same automated monitor. Elevated BP was defined as SBP and DBP values at or above the 90th percentile for sex, age, and height [1].

Total daily PA was assessed using the ActiGraphTM GT1M accelerometer (Pensacola, FL, USA), which records PA intensity levels; the higher the recorded sum, the greater the intensity [21]. Participants wore the accelerometer on their hip for seven consecutive days (Monday through Sunday), ensuring a minimum of 10 h of wear time each day. For analysis, valid data were required for at least 4 days [22]. Periods of nonwear were defined as intervals of 60 consecutive minutes with zero counts [23]. Data were processed, cleaned, and analyzed using the ActilifeTM software. MVPA was categorized using a threshold of ≥60 min/day vs. <60 min/day. Sedentary behavior was classified based on its distribution, with individuals categorized as above or below the 75th percentile of observed sedentary time.

Data normality was assessed using the Kolmogorov–Smirnov test. Differences between sexes were analyzed using Student’s *t*-test, Mann–Whitney test, or Chi-square test, depending on the data distribution. Poisson regression with robust error variance was employed to calculate prevalence ratios (PR) and 95% confidence intervals (95% CI) for elevated BP across age groups, BMI, WHtR, sedentary behavior, and MVPA. All calculations were performed using SPSS software (version 24.0; SPSSTM Inc., Chicago, IL, USA), with the significance level set at 5%.

## 3. Results

A total of 2901 children and adolescents were included in this study (Table 1). The results showed that 31.9% of participants were living with overweight or obesity, while 65.1% and 66.1% met the criteria for a healthy WHtR and MVPA, respectively. The chi-square test for trend highlighted a significant linear association between BP and age groups, showing a consistent increase in elevated BP values across childhood and adolescence. Additionally, a significant association was found between BMI, WHtR, sedentary behavior, MVPA, and BP.

Table 2 presents the PR for elevated BP in children using unadjusted (crude) and adjusted (multivariable) models according to age group, BMI, WHtR, sedentary behavior, and MVPA. In the crude analysis, middle and high schoolers demonstrated a PR of 2.1 and 1.9 times higher, respectively, for elevated BP than preschoolers. However, in the adjusted analysis, statistical significance remained only for middle schoolers, with a PR 1.8 times higher for elevated BP than for preschoolers.

Both the crude and multivariable analyses showed that the PR for elevated BP was 1.49 times higher in children and adolescents classified as overweight/obese and 1.4 times higher in those with a WHtR ≥ 0.50. These associations persisted after adjusting for other covariates. Interestingly, in the crude analysis, children and adolescents in the ≥75th percentile for sedentary behavior showed a PR 2.4 times lower for elevated BP than those below the 75th percentile, although this association was not confirmed in the adjusted model. The PR for high BP in children and adolescents was 2.79 (crude model) and 1.63 (adjusted model) times higher, respectively, among those not meeting the recommended 60 min of MVPA, even after adjusting for other covariates.

## 4. Discussion

This study examined the associations between physical activity levels, adiposity, sedentary behavior, advancing age, and the risk of elevated BP in a large sample of Portuguese children and adolescents. The findings indicated that elevated BP was significantly associated with older age, higher BMI, increased WHtR, insufficient physical activity, and sedentary behavior, to a lesser extent.

The significant and progressive increase in elevated BP prevalence with age observed here aligns with previous research by Song et al. [4] and Dong et al. [3], who also documented an increase in BP during adolescence. This increase is likely influenced by physiological changes such as hormonal shifts and lifestyle factors associated with puberty, including alterations in physical activity and dietary habits. Specifically, our study found that middle schoolers had a 1.8 times higher risk of elevated BP than preschoolers, underscoring adolescence as the critical period for the onset of elevated BP. It is important to emphasize that the elevated BP prevalence in our study was measured during a single visit, which may contribute to higher prevalence rates compared to studies like Dong et al. [3], where hypertension diagnosis required measurements across three separate visits. Evidence indicates that the prevalence of childhood hypertension tends to decrease as the number of measurement visits increases, which may account for the relatively higher prevalence estimates in this study [24]. This methodological difference underscores the need for caution when comparing prevalence estimates across studies. Furthermore, the tracking of elevated BP from childhood and adolescence into adulthood, as noted by Flynn et al. [1], reinforces the importance of early detection and intervention to mitigate long-term cardiovascular risk.

A strong association was observed between elevated BP and higher BMI and WHtR, with a 1.49 times higher PR for those classified as overweight or obese and a 1.4 times higher PR for those with a WHtR ≥ 0.50. This finding corroborates the work of Wang et al. [7], who showed a significantly increased risk of hypertension in children and adolescents with overweight or obesity. Our use of WHtR as an indicator of central obesity aligns with the findings of Sorof et al. [5] and Falaschetti et al. [6], which supports WHtR as a valuable marker for assessing cardiovascular risk in pediatric populations. The nearly doubled prevalence of elevated BP among children with higher adiposity emphasizes the importance of routine anthropometric screenings during pediatric visits, as recommended by health organizations such as the American Academy of Pediatrics and the American Heart Association [25]. These assessments could serve as early intervention points for identifying children at a higher risk of hypertension and other cardiometabolic issues.

The study’s finding of an inverse relationship between MVPA and elevated BP highlights the protective effects of physical activity, with only 15.3% of children meeting MVPA recommendations with elevated BP compared to 42.7% of those who did not meet these recommendations. This aligns with the findings of Raitakari et al. [9] and Guthold et al. [10], who emphasized the long-term cardiovascular benefits of sustained physical activity, especially in preventing and managing cardiovascular risk factors from an early age.

However, when examining sedentary behavior, it was found that its association with BP became statistically insignificant after adjustments, suggesting that a lack of regular physical activity may be a more critical factor for BP regulation than sedentary behavior alone. This distinction is essential, as sedentary behavior and physical inactivity are not equivalent concepts: sedentary behavior refers to time spent in low-energy activities, such as watching TV or using electronic devices, whereas physical inactivity means an overall absence of activities that significantly increase energy expenditure, such as MVPA [26].

In practice, replacing sedentary time with MVPA appears to be a more effective strategy for managing BP in children than merely reducing sedentary behavior. The literature supports this perspective: systematic reviews show that substituting sedentary behavior with MVPA is consistently associated with cardiovascular benefits, including reductions in both systolic and diastolic BP [27,28].

Therefore, the findings from this study suggest that interventions aimed at lowering BP in pediatric populations should prioritize increasing MVPA levels, targeting at least 60 min per day, rather than focusing solely on reducing sedentary time. This emphasis on substituting sedentary behavior with more active time aligns with public health guidelines [29,30], which advocate regular physical activity as a core strategy for the prevention and management of pediatric hypertension.

The strengths of this study include its large and diverse sample size, which bolsters the generalizability of its findings, and its comprehensive examination of multiple lifestyle factors, offering a robust perspective on their relationships with elevated BP in children and adolescents. Additionally, a key strength of this study is the use of objective measures of PA and sedentary behavior, which provide more reliable and accurate data compared to self-reported measures. However, this study had several limitations. BP was measured at a single time point, which could have overestimated the prevalence of elevated BP, as indicated by Flynn et al. [1]. Additionally, owing to data constraints, this study could not account for other critical confounders, including dietary factors, genetic predispositions, and other lifestyle behaviors that could affect BP outcomes, as noted in reviews by Almahmoud et al. [2] and Lona et al. [8]. Another limitation is the lack of a formal sample size calculation, as the sample was determined based on feasibility and data availability rather than a priori statistical power analysis. Furthermore, the data were collected between 2010 and 2015, and potential changes in lifestyle behaviors and environmental factors over time may limit the current applicability of the findings. These limitations highlight the need for future longitudinal studies to confirm these findings and to further explore the causal pathways linking lifestyle behaviors to elevated BP in this population. Such studies could provide valuable insights into the role of specific lifestyle factors and the efficacy of early interventions in preventing hypertension in children and adolescents.

## 5. Conclusions

This study demonstrates that advancing age, higher BMI, central obesity (as measured by WHtR), and insufficient physical activity are significantly associated with elevated BP in a large cohort of Portuguese children and adolescents. These findings underscore the critical importance of the early identification and management of these risk factors in the prevention of pediatric hypertension.

Given that elevated BP in youth can be tracked into adulthood, these results suggest that implementing routine screenings for BMI, WHtR, and physical activity levels could be invaluable for early detection and intervention efforts. Public health strategies focused on promoting healthier lifestyles, encouraging physical activity, and reducing central obesity in children may play a vital role in reducing the long-term burden of hypertension and its associated cardiovascular diseases. Early intervention not only supports healthier childhood development but also contributes to better cardiovascular health outcomes later in life.

## Figures and Tables

**Table 1 ijerph-22-00020-t001:** Distribution of Participant Characteristics According to Blood Pressure Profile (n = 2901).

Characteristic	Normal BP		Elevated BP		*p*-Value
	n	%	n	%	
Age groups					
Preschoolers (2–5 years)	812	80.2	200	19.8	<0.001 ^a^
Primary-Schoolers (6–10 years)	257	76.0	81	24.0	
Middle-Schoolers (11–14 years)	718	57.6	529	42.4	
High-Schoolers (15–18 years)	185	60.9	119	39.1	
BMI					
Eutrophic	1472	74.5	503	25.5	<0.001 ^b^
Overweight/obesity	499	53.9	426	46.1	
WHtR					
<0.49	1300	70.7	540	29.3	<0.001 ^b^
≥0.50	614	62.1	374	37.9	
Sedentary Behavior					
<75th percentile	770	64.5	424	35.5	<0.001 ^b^
≥75th percentile	348	73.1	128	26.9	
MVPA					
≥60 min/day	747	84.7	135	15.3	<0.001 ^b^
<60 min/day	259	57.3	193	42.7	

^a^ Chi-square test for linear trend; ^b^ Chi-square. BMI: body mass index; WHtR: waist-to-height ratio; MVPA: moderate-to-vigorous physical activity.

**Table 2 ijerph-22-00020-t002:** Prevalence Ratios for Elevated Blood Pressure in Children: Crude and Multivariable Analyses by Demographic and Lifestyle Factors (n = 2901).

Variables	Elevated BP (>90th Percentile Based on Age, Sex, and Height Percentiles)			
	Crude Analysis		Multivariable Analysis	
	PR (CI 95%)	Wald (*p*-Value)	PR (CI 95%)	Wald (*p*-Value)
Sex				
Male (n = 1469)	1	1.535 (0.215)	-	-
Female (n = 1432)	1.069 (0.962–1.189)			
Age groups				
Preschoolers (n = 1012)	1	132.968 (<0.001)	1	10.638 (0.005)
Primary-Schoolers (n = 338)	1.213 (0.966–1.521)		1.308 (0.950–1.802)	
Middle-Schoolers (n = 1247)	2.147 (1.866–2.469) *		1.844 (1.276–2.665) *	
High-Schoolers (n = 304)	1.981 (1.643–2.389) *		-	
BMI				
Eutrophic (n = 1975)	1	127.694 (<0.001)	1	9.382 (0.002)
Overweight/obesity (n = 925)	1.808 (1.632–2.004)		1.490 (1.154–1.922)	
WHtR				
<0.49 (n = 1840)	1	21.812 (<0.001)	1	6.755 (0.009)
≥0.50 (n = 988)	1.290 (1.159–1.435)		1.393 (1.085–1.789)	
Sedentary Behavior				
<75th percentile (n = 1194)	1	10.690 (0.001)	1	0.272 (0.602)
≥75th percentile (n = 476)	0.757 (0.641–0.895)		0.917 (0.663–1.270)	
MVPA				
≥60 min/day (n = 882)	1	127.694 (<0.001)	1	11.203 (<0.001)
<60 min/day (n = 452)	2.790 (2.311–3.368)		1.636 (1.226–2.182)	

* Denotes statistically significant results with *p* < 0.05. PR: prevalence ratio; BMI: body mass index; WHtR: waist-to-height ratio; MVPA: moderate-to-vigorous physical activity.

## Data Availability

Data supporting the findings of this study are available upon reasonable request from the corresponding author.

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
