# Peer review of "Relationship Among Body Mass Index, Physical Activity, Sedentary Behavior, and Blood Pressure in Portuguese Children and Adolescents: A Cross-Sectional Study"

_ijerph, 2024, doi:10.3390/ijerph22010020_

Round 1
Reviewer 1 Report
Comments and Suggestions for Authors
The article aims to identify the factors associated with elevated blood pressure in children and adolescents, highlighting the importance of monitoring blood pressure from an early age. The study is well organized and presents its results clearly, aligning them with the existing literature. Furthermore, it interprets the findings clearly and suggests future interventions aligned with general public health guidelines.
The abstract provides an overview of the content of the manuscript. It adequately summarizes the study objectives and findings, as well as highlighting the need for follow-up at early ages to prevent hypertension.
The introduction presents a general background on all parameters associated with blood pressure. However, there is a lack of agreement between the stated objectives and the introduction. The objectives mention nutritional status and adiposity, but these aspects are not addressed in the introductory section.
The material and methods section covers the research design, ethics, variables, and instruments. However, it is essential to provide more information on the study participants, how they were recruited, and the inclusion/exclusion criteria applied.
The results section is well organized and presents the data using tables. The results are well structured and a brief description is added to facilitate understanding.
The discussion adequately interprets the results of the study and aligns them with the existing literature. In addition, the limitations of the study and a possible future line of research are raised. However, it is necessary to cite the general public health guidelines mentioned in line 200.
In relation to the references section, it is necessary to review and correct the format of the bibliographic references to ensure that the editorial standards are respected.
Author Response
Reviewer 1
The article aims to identify the factors associated with elevated blood pressure in children and adolescents, highlighting the importance of monitoring blood pressure from an early age. The study is well organized and presents its results clearly, aligning them with the existing literature. Furthermore, it interprets the findings clearly and suggests future interventions aligned with general public health guidelines.
Answer: Thank you for your feedback. We are pleased the study’s organization, clarity, and alignment with the literature meet expectations.
The abstract provides an overview of the content of the manuscript. It adequately summarizes the study objectives and findings, as well as highlighting the need for follow-up at early ages to prevent hypertension.
Answer: Thank you for your comment. We are pleased that the abstract effectively summarizes the study's objectives, findings, and the importance of early follow-up to prevent hypertension.
The introduction presents a general background on all parameters associated with blood pressure. However, there is a lack of agreement between the stated objectives and the introduction. The objectives mention nutritional status and adiposity, but these aspects are not addressed in the introductory section.
Answer: Thank you for your comment. We have revised the introduction to explicitly address nutritional status and adiposity, ensuring alignment with the study's stated objectives.
The material and methods section covers the research design, ethics, variables, and instruments. However, it is essential to provide more information on the study participants, how they were recruited, and the inclusion/exclusion criteria applied.
Answer: Thank you for your comment. We have revised the Material and Methods section to include additional details about the study participants, recruitment process, and the inclusion/exclusion criteria applied.
The results section is well organized and presents the data using tables. The results are well structured and a brief description is added to facilitate understanding.
Answer: Thank you for your comment. I appreciate your positive feedback regarding the organization and clarity of the results section
The discussion adequately interprets the results of the study and aligns them with the existing literature. In addition, the limitations of the study and a possible future line of research are raised. However, it is necessary to cite the general public health guidelines mentioned in line 200.
Answer: Thank you for the comment. We have now cited the relevant public health guidelines to support our statement.
In relation to the references section, it is necessary to review and correct the format of the bibliographic references to ensure that the editorial standards are respected.
Answer: We appreciate the reviewer’s comment regarding the references section. A thorough revision has been conducted to ensure that all bibliographic references comply with the Reference List and Citations Style Guide for MDPI Journals.
Reviewer 2 Report
Comments and Suggestions for Authors
The topic of this manuscript is very up-to-date and readable. Considering that the relationship between physical activity and other mentioned indicators and diseases related to blood pressure is of great importance. It seemed that this manuscript could also show this importance more than before. Therefore, considering the good report in this manuscript, I recommend this manuscript for publication. Only a few things to improve the quality of the manuscript are mentioned below.
The introduction is well written. However, it is suggested that newer studies be added to the introduction and that the research hypotheses be mentioned at the end of the introduction.
How was the sample size calculated? Was statistical software used?
Was the data from 2010 to 2014? Considering the length of time and possible differences, it is suggested that this be mentioned in the limitations section.
If possible, a bar chart should be used to compare groups. Also, the effect size should be reported for all comparisons.
Given that there are two types of objective and subjective measurements for physical activity (of course, you used objective measurements, which is a strength of your study), it would be better to address this issue further in both the introduction and the discussion.
Given that BMI in athletes cannot be a correct indicator (because athletes mainly have more muscle mass, which leads to their weight gain), it would be better to mention this issue in the discussion and limitations section. Did you ask the participants whether they were athletes or not?
Author Response
Reviewer 2
The topic of this manuscript is very up-to-date and readable. Considering that the relationship between physical activity and other mentioned indicators and diseases related to blood pressure is of great importance. It seemed that this manuscript could also show this importance more than before. Therefore, considering the good report in this manuscript, I recommend this manuscript for publication. Only a few things to improve the quality of the manuscript are mentioned below.
Answer: We sincerely thank the reviewer for their positive feedback and recommendation for publication. We appreciate the constructive suggestions provided to further improve the quality of the manuscript.
The introduction is well written. However, it is suggested that newer studies be added to the introduction and that the research hypotheses be mentioned at the end of the introduction.
Answer: Thank you for your suggestion. I have added recent studies to the introduction and included the research hypotheses at the end, as requested.
How was the sample size calculated? Was statistical software used?
Answer: Thank you for your valuable comment. The issue regarding the sample size calculation has been addressed and included in the study's limitations section.
Was the data from 2010 to 2014? Considering the length of time and possible differences, it is suggested that this be mentioned in the limitations section.
Answer: Thank you for your comment. We have incorporated this point into the limitations section as suggested.
If possible, a bar chart should be used to compare groups. Also, the effect size should be reported for all comparisons.
Answer: Thank you for your comment. While we appreciate the suggestion to include a bar chart for group comparisons and report effect sizes, it is important to note that comparing groups was not the primary objective of this study.
Given that there are two types of objective and subjective measurements for physical activity (of course, you used objective measurements, which is a strength of your study), it would be better to address this issue further in both the introduction and the discussion.
Answer: Thank you for your comment. The distinction between objective and subjective measurements of physical activity is well-established in the literature. Given its consolidation in existing research, we chose to focus on emphasizing the strength of using objective measurements in our study, which was mentioned in the discussion section. We did not address this issue in the introduction, as the primary focus was on presenting the overall context and objectives of the study.
Given that BMI in athletes cannot be a correct indicator (because athletes mainly have more muscle mass, which leads to their weight gain), it would be better to mention this issue in the discussion and limitations section. Did you ask the participants whether they were athletes or not?
Answer: Thank you for your insightful comment. It is important to note that the study sample consisted exclusively of children and adolescents, and we did not specifically inquire about participants' involvement in athletic activities. While we recognize that athletes may have different body compositions due to increased muscle mass, which could affect BMI measurements, we have not included this as a limitation in the study, as the focus was on a general pediatric population.
Reviewer 3 Report
Comments and Suggestions for Authors
The present study is a cross-sectional design of over 2900 Portuguese children and adolescents aged 2–18 16 years. They gathered by protocol: BP, body mass index (BMI), waist-to-height ratio (WtHR), physical activity, and sedentary behavior measurements. Elevated BP was defined as a BP above the 90th percentile for age, sex, and height for four separate age groups. Multivariable analyses assessed the prevalence ratios (PR) of elevated BP across age groups, BMI, WtHR, physical activity, and sedentary behavior. The findings indicated that elevated BP was significantly increased with older age , higher BMI and increased WtHR, insufficient physical activity, and sedentary behavior, to a lesser extent.
An important finding was the comparison of amount of sedentary behavior compared to the effect of time involved in significant increased energy expenditure measured by (moderate versus vigorous activity) : MVPA duration. This was most significantly associated with BP and to a lesser extent duration sedentary activity.
Question and edits include: Credits on cover page are not in English and need to be translated: Departamento de Ciências do Esporte, Universidade Federal do Triângulo Mineiro, Uberaba, MG, Brazil; 7 2 Centro de Investigação em Atividade Física, Saúde e Lazer, Faculdade de Desporto da Universidade do 8 Porto e Laboratório para a Investigação Integrativa e Translacional em Saúde Populacional (ITR), Porto, 9 Portugal 10 3 Laboratório de Atividade Física e Promoção da Saúde, Instituto de Educação Física e Desportos, 11 Universidade do Estado do Rio de Janeiro, Rio de Janeiro, RJ, Brazil
Some terms need to be defined :
They include: Prevalence Ratio ( PR) please describe , and your definition of BMI :Eutrophic 1, Overweight and obesity by your measures, in lbs.
Also when you refer to prevalence of HBP which is typically >90 mm hg , can you put this number into the context of explaining >90% norms for each of the four age groups? It is difficult to square this in terms of your introduction in which you cite 4% of children have hypertension from previous studies, how many of your 2900 children had hypertension using the standard definition versus the " norms " you talk about.?
This will help immeasurably to help the reader to determine where your population is in relation to the overall literature.
Author Response
Reviewer 3:
The present study is a cross-sectional design of over 2900 Portuguese children and adolescents aged 2–18 16 years. They gathered by protocol: BP, body mass index (BMI), waist-to-height ratio (WtHR), physical activity, and sedentary behavior measurements. Elevated BP was defined as a BP above the 90th percentile for age, sex, and height for four separate age groups. Multivariable analyses assessed the prevalence ratios (PR) of elevated BP across age groups, BMI, WtHR, physical activity, and sedentary behavior. The findings indicated that elevated BP was significantly increased with older age , higher BMI and increased WtHR, insufficient physical activity, and sedentary behavior, to a lesser extent.
An important finding was the comparison of amount of sedentary behavior compared to the effect of time involved in significant increased energy expenditure measured by (moderate versus vigorous activity) : MVPA duration. This was most significantly associated with BP and to a lesser extent duration sedentary activity.
Answer: Thank you for your thorough review of our study. We appreciate the time and effort you dedicated to providing valuable feedback.
Question and edits include: Credits on cover page are not in English and need to be translated: Departamento de Ciências do Esporte, Universidade Federal do Triângulo Mineiro, Uberaba, MG, Brazil; 7 2 Centro de Investigação em Atividade Física, Saúde e Lazer, Faculdade de Desporto da Universidade do 8 Porto e Laboratório para a Investigação Integrativa e Translacional em Saúde Populacional (ITR), Porto, 9 Portugal 10 3 Laboratório de Atividade Física e Promoção da Saúde, Instituto de Educação Física e Desportos, 11 Universidade do Estado do Rio de Janeiro, Rio de Janeiro, RJ, Brazil
Answer: Thank you for your comment. The requested changes have been made.
Some terms need to be defined :
They include: Prevalence Ratio ( PR) please describe , and your definition of BMI :Eutrophic 1, Overweight and obesity by your measures, in lbs.
Answer: We appreciate the reviewer’s suggestion and have addressed the requested clarifications. The definition of Prevalence Ratio (PR) has been included in the methods section. Regarding BMI classification, we clarified that BMI was maintained in kg/m², as it is derived from the WHO growth reference curves for school-aged children and adolescents from de Onis M, Onyango AW, Borghi E, Siyam A, Nishida C, Siekmann J. Development of a WHO growth reference for school-aged children and adolescents. Bull World Health Organ. 2007;85(9):660-667. doi:10.2471/blt.07.043497. Additionally, the classification of BMI into normal weight, overweight, and obesity has been explicitly described in the methods section, following the WHO growth reference standards.
Also when you refer to prevalence of HBP which is typically >90 mm hg , can you put this number into the context of explaining >90% norms for each of the four age groups? It is difficult to square this in terms of your introduction in which you cite 4% of children have hypertension from previous studies, how many of your 2900 children had hypertension using the standard definition versus the " norms " you talk about.? This will help immeasurably to help the reader to determine where your population is in relation to the overall literature.
Answer: We appreciate the reviewer’s observations and the opportunity to clarify. Our study focused on presenting data for elevated blood pressure (EBP), which was defined using the normative BP tables based on normal-weight children as normal (50th percentile), elevated BP (>90th percentile), stage 1 hypertension (≥95th percentile), and stage 2 hypertension (≥95th percentile + 12 mm Hg). The prevalences of elevated BP observed in our study were 19.8% for preschoolers (2–5 years), 24.0% for primary-schoolers (6–10 years), 42.4% for middle-schoolers (11–14 years), and 39.1% for high-schoolers (15–18 years). It is important to emphasize that these prevalences reflect elevated BP rather than hypertension, which requires SBP and/or DBP ≥95th percentile (for age, sex, and height) on at least three separate occasions. The relatively higher prevalences reported in our study compared to previous studies can be explained by several factors, including the fact that BP measurements were taken during a single visit, whereas the diagnosis of hypertension necessitates repeated measurements across multiple visits. We have clarified these distinctions in the manuscript to avoid potential misunderstandings regarding the classification and prevalence of elevated BP versus hypertension.
Round 2
Reviewer 3 Report
Comments and Suggestions for Authors
The authors have addressed my concerns in this new revision and I find it acceptable, It is difficult to accept sedentary behavior as not being associated with a higher risk of elevated BP but the authors stated that this not inactivity but time spent in relaxing activities independent of vigorous activity that was found to incur protection from elevations in BP.eg. the finding of those not meeting the recommended 60 minutes of moderate-to-vigorous physical activity (MVPA) per day showed a 1.63 times higher risk of elevated BP, whereas the association between sedentary behavior and BP was not significant after adjustment.